# Lumican Inhibits In Vivo Melanoma Metastasis by Altering Matrix-Effectors and Invadopodia Markers

**DOI:** 10.3390/cells10040841

**Published:** 2021-04-08

**Authors:** Konstantina Karamanou, Marco Franchi, Isabelle Proult, Romain Rivet, Demitrios Vynios, Stéphane Brézillon

**Affiliations:** 1CNRS UMR 7369, Matrice Extracellulaire et Dynamique Cellulaire, 51100 Reims, France; kon.karamanou@gmail.com (K.K.); isabelle.proult@univ-reims.fr (I.P.); romain.rivet@univ-reims.fr (R.R.); 2Laboratoire de Biochimie Médicale et Biologie Moléculaire, Université de Reims Champagne Ardenne, 51095 Reims, France; 3Biochemistry, Biochemical Analysis & Matrix Pathobiology Research Group, Laboratory of Biochemistry, Department of Chemistry, University of Patras, 26501 Patras, Greece; vynios@chemistry.upatras.gr; 4Department for Life Quality Studies, University of Bologna, 47922 Rimini, Italy; marco.franchi3@unibo.it

**Keywords:** lumican, melanoma, lung metastasis, invadopodia, epithelial-to-mesenchymal transition, snail

## Abstract

It was reported that lumican inhibits the activity of metalloproteinase MMP-14 and melanoma cell migration in vitro and in vivo. Moreover, Snail triggers epithelial-to-mesenchymal transition and the metastatic potential of cancer cells. Therefore, the aim of this study was to examine the effect of lumican on Mock and Snail overexpressing melanoma B16F1 cells in vivo. Lung metastasis was analyzed after intravenous injections of Mock-B16F1 and Snail-B16F1 cells in Lum^+/+^ and Lum^−/−^ mice. At day 14, mice were sacrificed, and lungs were collected. The number of lung metastatic nodules was significantly higher in mice injected with Snail-B16F1 cells as compared to mice injected with Mock-B16F1 cells confirming the pro-metastatic effect of Snail. This effect was stronger in Lum^−/−^ mice as compared to Lum^+/+^, suggesting that endogenous lumican of wild-type mice significantly inhibits metastasis to lungs. Scanning electron and confocal microscopy investigations demonstrated that lumican inhibits the development of elongated cancer cell phenotypes which are known to develop invadopodia releasing MMPs. Moreover, lumican was shown to affect the expression of cyclin D1, cortactin, vinculin, hyaluronan synthase 2, heparanase, MMP-14 and the phosphorylation of FAK, AKT, p130 Cas and GSK3α/β. Altogether, these data demonstrated that lumican significantly inhibits lung metastasis in vivo, as well as cell invasion in vitro, suggesting that a lumican-based strategy targeting Snail-induced metastasis could be useful for melanoma treatment.

## 1. Introduction

The incidence of melanoma is increasing annually at a high rate. Although melanoma represents only 5% percent of skin cancers, it is an aggressive neoplasm and it can rapidly become life-threatening once it has metastasized. Malignant melanoma results in high mortality rates with the higher potential of dissemination among the malignancies. It is responsible for most skin cancer-associated deaths with over 200,000 diagnosed melanoma cases each year worldwide, leading to 55,000 deaths [1]. It preferentially metastasizes to distant lymph nodes and to organs such as lungs (18–36%), liver (14–20%) and brain (12–20%) [2]. Notably, pulmonary metastatic melanoma is often asymptomatic. Therefore, early diagnosis and better prevention are important. While environmental factors, lifestyle and phenotypic and genetic susceptibility contribute to the initiation of melanoma, etiology is complex, multifactorial and heterogeneous. For instance, UV radiation is one of the most considerable environmental risk factors, displaying a mutagenic role in melanoma pathogenesis [2]. Moreover, phenotypic risk factors, including light pigmentation, tendency to sunburn, inability to tan, misshaped lesions with several protrusions and a big number of melanocytic nevi play a major role in the metastatic dissemination of melanoma [3,4].

In malignant melanoma, the role of cyclins and especially cyclin D1 and D3 are well described. It is reported that abnormal cyclin D3 expression is correlated with poor clinical outcome, as well as high rate of metastatic melanoma lesions in superficial melanoma [5]. Cyclin D1 is responsible for promoting cell cycle progression through the G_1_-S phase by encoding the regulatory subunit of a holoenzyme, which phosphorylates and inactivates the retinoblastoma protein. Overexpression of cyclin D1 is mainly related to human tumorigenesis and metastasis progression, eventually affecting the development of several human cancer types, such as parathyroid adenoma, breast cancer, colon cancer, prostate cancer, lymphoma and melanoma [6]. It is worth noticing that NRAS or BRAF mutation are frequent and associated with melanoma subtypes together with cyclin D1 and other factors [7]. Despite the advances in the treatment of malignant melanoma with BRAF/MEK targeted agents and immunotherapy, a poor prognosis for advanced disease is noted and therefore cyclin-dependent kinase inhibition either as single agent or in combination with established treatment are under clinical trials [8].

Transcription factor Snail is reported to potentially induce epithelial-to-mesenchymal transition (EMT) endowing cells with increased migratory and invasive properties [9]. During melanoma progression, cancer cell proliferation is disrupted because of Snail–mediated downregulation of E-cadherin [10]. Moreover, the inhibition of Snail-triggered EMT resulted in melanoma tumor suppression [11].

Melanoma metastasis is characterized by an invasion of the tumor cells to the nearest tissues and a formation of secondary sites of tumor growth after colonization of distant tissues. A crucial step for the initiation of melanoma metastasis is the degradation of key components of the basement membrane and macromolecules of the extracellular matrix (ECM) by matrix metalloproteinases (MMPs). Cancer cells have been reported to express high levels of MMP-14 (known also as MT1-MMP) especially in the cell protrusions and extracellular vesicles (EVs) [12,13]. Cancer cell-derived EVs are increasingly being recognized as genuine invasive structures as they contribute to many aspects of invasion. A recent report indicates a role of the actin cytoskeleton in the mechanisms underlying EV biogenesis or release. Indeed, Els Beghein et al. have demonstrated a role of the cortactin in EV release [12]. A contribution of this protein in endosomal trafficking was shown to be a crucial step in EV biogenesis. EVs are preferentially released at invadopodia, the latter being actin rich invasive cell protrusions where cortactin plays a regulatory role. Accordingly, EVs are enriched with invadopodial proteins such as the MMP-14 and exert gelatinolytic activity [12]. Thus, cortactin plays key roles in EV release by regulating endosomal trafficking or invadopodia formation and function.

Hyaluronan (HA), a linear glycosaminoglycan, with extremely complex functions, plays crucial roles in the development and progression of malignant tumors [14]. The contribution of HA to tumorigenesis relies on its metabolism, depending on the hyaluronan synthases (HASs) by the mean of the upregulation of HAS2 and HA overexpression as well as by hyaluronidases (HYALs) that degrade HA to various molecular sizes. It has been demonstrated that UDP-glucose 6-dehydrogenase regulates HA production and promotes breast cancer progression [15]. Moreover, binding of HA to its cellular receptors, mainly CD44 and RHAMM, leads to the activation of several tumorigenic signaling pathways and regulation of cellular functions. It is worth noticing that decreased HA amount is related with elevating content of degrading enzyme HYAL2, and the decreased expression of HAS1 and HAS2 in invasive melanoma [16,17].

Heparanase as the only mammalian endoglycosidase, which is able to degrade heparan sulfate side chains of proteoglycans (PGs), plays a crucial role in the remodeling of ECM, and is implicated in the EMT cellular invasion, related to angiogenesis, inflammation and metastasis. Heparanase expression is elevated in several types of tumors and this high expression pattern is mostly related to aggressive type of disease, poor prognosis, as well as increased tumor metastasis [18,19,20].

Melanoma metastasis correlates with the degradation of macromolecules of the ECM. Among them, PGs are critically implicated in several pathophysiological processes, such as cancer [14]. Small leucin-rich PGs (SLRPs) are secreted in the ECM and are able to interact with matrix effectors, such as cytokines, growth factors and cell surface proteins [21]. These interactions endow SLRPs with important cell functional properties, such as regulation of migration, invasion, angiogenesis and metastasis. Lumican is a Class II SLRP, with high molecular heterogeneity due to its glycosylation which is tissue-dependent [22,23]. Structurally, its core protein consists of three major domains: a negatively charged N-terminal domain, containing tyrosine sulfates and cysteine residues, a central part, which is characterized by nine leucine rich repeats, and a C- terminal domain of 66 amino acids [22,23].

According to the tissue, lumican structure and expression differ qualitatively and quantitatively, respectively. For instance, lumican mRNA exhibits very low expression in liver and sclera, and is almost undetectable in brain and sternum [23]. It has been shown that lumican delays melanoma growth in mice and plays regulatory roles in functional properties and invadopodia formation in breast cancer cells [24].

Biologically, lumican is able to regulate various physiological processes, like collagen fibrillo-genesis, cell proliferation, adhesion, migration and invasion [21,25,26]. Moreover, the anticancer effect of lumican was reported in numerous publications [13,24,25,26,27,28,29,30,31,32,33,34,35]. The former studies [28,29] utilized as a melanoma model, B16F1 cells transfected with the human lumican cDNA injected subcutaneously in wild-type C57BL/6J mice. Lumican was shown to inhibit the development of the melanoma primary tumor as well as lung metastatic nodules development [28,29]. More recently, another mouse melanoma model was investigated in which melanoma B16F1 cells were transfected with human Snail cDNA and then injected subcutaneously in wild-type and Lum^−/−^ C57BL/6J mice. In this model, endogenous lumican was demonstrated to inhibit the primary melanoma tumor development, while Snail overexpression was shown to induce EMT and the metastatic potential of melanoma cells [11,13]. The potential anti-metastatic role of lumican in melanoma by inhibiting the membrane type matrix metalloproteinase MMP-14 activity and melanoma cell migration in vitro was also reported [13,33,34].

Collectively, it has been previously reported that lumican, among other functions, effectively regulates cell functional properties, expression of ECM effectors, EMT, invadopodia markers, morphology of invading breast cancer cells, as well as MMP-14 and cell migration in melanoma cells [14,24,33,34]. Notably, lumican was shown to be a strong endogenous inhibitor of tumor growth [34]. Therefore, the aim of the present study was to investigate the development of metastatic melanoma nodules in the lung following intravenous injection of Snail-transfected B16F1 cells in lumican deficient and wild type C57BL/6J mice as well as to analyze the effect of lumican on Mock and Snail overexpressing B16F1 cells in vivo and in vitro focusing on invadopodia formation and the associated signaling pathways.

## 2. Materials and Methods

### 2.1. Extracellular Matrix PROTEINS

Recombinant human lumican (57 kDa) was purchased from R&D Systems (2846-LU-050, R&D Systems, Minneapolis, MN, USA). Type I collagen was extracted from rat tail tendon, as previously described [30]. Recombinant human (rh) MMP-14 (catalytic domain, aa 89-265, EC 3.4.24.-) was purchased from Merck Millipore^®®^ (Nottingham, UK).

### 2.2. Animal Care

Heterozygous Lum^+/−^ mice (C57BL/6J) were kindly provided by Pr. Shukti Chakravarti. After multiple breeding, homozygous Lum^−/−^ mice were obtained. The distinction between homozygous Lum^−/−^ mice and their wild-type (Lum^+/+^) littermates was performed by PCR genotyping. Two pairs of primers, one pair internally from E2 (forward primer, 5′-CCTGAGGAATAACCAAATCGACC-3′ and reverse primer, 5′-AGGCAGCTTGCTCATCTGAATTGA-3′), the second pair from the pGKneo cassette (forward primer 5′-CATTCGACCACCAAGCGAAAC-3′ and reverse primer 5′-AGCTCTTCAGCAATATCACGGG-3′) were used to amplify a 380-bp and a 300-bp product from the wild type and mutant allele, respectively. All mice procedures conformed to the ethical rules of the University of Reims Champagne-Ardenne ethical committee and were approved by the URCA Animal facility platform (URCAnim, Reims EU0362), and the CNRS. This study was performed respecting the “The French Animal Welfare Act” and “The French Board for Animal Experiments”. Experiments were conducted under approval of the French “Ministère de l’Enseignement Supérieur et de la Recherche” (ethics committee n°C2EA-56) in compliance with the “Directive 2010/63/UE”, Protocol number: APAFIS#17470-2018110911091242V2.

### 2.3. Cell Culture and Transfection

B16F1 melanoma cells were transfected with the plasmids containing human SNAIL. The pcDNA 3.1-human SNAIL construct was obtained from Prof. Muh-Hwa Yang (Taipei Veterans General Hospital, Taiwan). B16F1 melanoma cells were grown up to 85% confluence in 1 g/L glucose DMEM Biosera LTD (Courtaboeuf CEDEX, France) in presence of 0.1% penicillin/streptomycin and 10% FCS at 37 °C with 5% CO_2_. The Amaxa Nucleofector X Unit (Lonza, Basel, Switzerland) was used for the transfection of cells with 5 μg DNA/10^6^ cells, according to the manufacturer’s instructions. Cells were cultured in medium with additional 200 μg/mL of Geneticin/G418 (Gibco/LifeTechnologies, Waltham, MA, USA). After two weeks of cell culture and refreshing selection media every two days, the colonies, which were well-separated, were selected. The culture of clones was scaled-up and SNAIL expression was verified by RT-PCR and Western blot analysis. Mock-B16F1 cells, transfected with an empty plasmid, cultured with media supplemented with G418 (200 μg/mL), were used as controls.

### 2.4. Lung Metastasis Analysis

Mock- or Snail-B16F1 cells (n = 2.5 × 10^5^ cells per mouse, 100 μL of cell suspension in DMEM) were injected in the tail vein of a total of 32 mice (Mock-B16F1 cells injected in WT-mice (n = 8), Mock-B16F1 cells injected in lumican deleted mice (n = 8), Snail-B16F1 cells injected in WT-mice (n = 8), Snail-B16F1 cells injected in lumican deleted-mice (n = 8)). At day 14, mice were sacrificed. The lungs were isolated from every animal and were rinsed in PBS. The number of nodules was calculated in lungs. Tissues were paraffin embedded and 5 μm thick sections were performed.

### 2.5. Histological and Immunohistochemical Analyses

Histological analyses of paraffin-embedded sections were performed after hematoxylin and eosin (HE) staining. Paraffin-embedded tissue sections from metastatic lungs were analyzed by immunohistochemistry for protein expression of cyclin D1. Immunohistochemistry was performed using the primary antibody of anti-cyclin D1 (see Appendix A). Biotin-labeled secondary antibodies, streptavidin-HRP, and DAB (3,3′Diaminobenzidine) detection system was provided by Abcam (Abcam, Paris, France). Negative controls were performed by omitting the primary antibody and by using control isotypes (Dako, Les Ulis, France) as negative controls. Cyclin-D1-positive cells were numbered in five random fields of the metastatic lung nodules area at 100× magnification. All quantitative analyses were performed using ImageJ software binarization and thresholding tools.

### 2.6. Scanning Electron Microscopy (SEM)

Mock-B16F1 and Snail-B16F1 cells were assessed by using the Boyden-chamber assay. Negative controls were also tested. “Isopore Membrane Filters” (Millipore^®®^, Milan, Italy) with pore size of 8.0 μm were used for the assays. Twenty-four filters were coated with Type I collagen from rat tendon. For this purpose, collagen was dissolved in 18 mM sterile acetic acid, in order to reach the concentration of 5 μg/100 μL. Initially, 100 μL from the collagen mixture were added in each well of the twenty-four filters trans-well plate, and the plate was incubated overnight at 4 °C. Another portion of 100 μL was added in each well in order to reach a total concentration of 10 μg/100 μL. For every cell line, 2 × 10^5^ cancer cells suspended in 800 μL serum-free medium were seeded on top of the gel in the upper chamber. Conditioned media were added in the lower chamber and were used as chemoattractant.

For each cell line, three filters were fixed after 24 h and thee additional filters were fixed after 48 h of cell culture (control samples). For each cell line, three different filters were treated with 100 nM of lumican for 24 h and other three for 48 h. All filters with cells seeded on the top were fixed in a Karnovsky’s solution for 20 min and then rinsed three times with 0.1% cacodylate buffer. Afterwards, samples were dehydrated with increasing concentrations of ethanol, and finally dehydrated with hexamethyl disilazane (Sigma-Aldrich Inc., Atlanta, GA, USA) for 15 min. All samples were placed on appropriate stubs, covered with a 5 nm palladium gold film (Emitech 550 sputter-coater, Ashford, UK), to be observed under a SEM (Philips 515, Eindhoven, The Netherlands) operating in secondary-electron mode.

### 2.7. Laser Scanning Microscopy

For Immunofluorescence/confocal microscopy experiments, cells were seeded on sterile glass coverslips in 24-well plates and grown to 50% confluence before treatment. Snail-B16F1 and Mock-B16F1 cells were seeded at 5 × 10^5^ cells/well. Cells were rinsed twice in PBS, fixed using 4% paraformaldehyde (in PBS buffer) pH 7.2 for 15 min at room temperature and washed three times with PBS-Tween buffer. After saturation with 3% BSA for 30 min at room temperature, cells were incubated overnight at 4 °C with the appropriate primary antibody. The references and the dilutions of the primary antibodies are displayed in the Appendix A. For the detection of actin cytoskeleton, cells were permeabilized with 0.1% Triton X-100 and incubated 1 h at room temperature with Alexa Fluor^®®^ 488-conjugated phalloidin at 1/200 dilution. Negative controls were also prepared after staining with mouse IgG. Slides were observed under confocal laser scanning microscope (Zeiss LSM 700, Zeiss, Marly le Roi, France).

### 2.8. Real Time RT-PCR

Total RNA of Mock- and Snail-overexpressing B16F1 cells, was isolated using RNeasy Plus Mini Kit (Qiagen, Courtaboeuf, France) according to the instructions of the Manufacturer. RNA quality was examined and determined on an Agilent 2100 Bioanalyzer (Agilent Technologies, Massy, France). Afterwards, reverse transcription was performed at 50 °C for 30 min using the Maxima First Strand TM cDNA synthesis kit (Thermo Scientific, Villebon and Yvette, France) with 1 μg of total RNA. The Instrument used was the Abi Prism Instrument (Fisher Scientific, Illkirch, France). The primers were purchased from Applied Biosystems (Applied Biosystems Fisher Scientific, Illkirch, France) and are displayed in the Appendix A. The relative quantifications and finally the results were obtained with the use of the ΔΔCt method. PCR assays were conducted at least three times in triplicates for each sample. Another Instrument used was the Mx3005P Thermocycler (SYBR Green) and the primers were purchased from Abi Taqman, displayed in the Appendix A. The relative quantifications and finally the results were obtained with the use of ΔΔCt method. PCR assays were conducted at least three times in triplicates for each sample. All primer sequences and sizes of the PCR product of each targeted gene are presented in Appendix A. The specificity of PCR amplification products was assessed by dissociation melting-curve analysis. After the reaction was completed, Ct value was calculated from the amplification plots. The standard curves were generated with serially diluted solutions (1/5–1/3125) of cDNA from B16F1 cells. Each sample was normalized to a housekeeping gene transcript either β-Actin or glyceraldehyde-3-phosphate dehydrogenase (GAPDH).

### 2.9. Western Immunoblotting

Total cell proteins were prepared from cell monolayers after being washed twice with PBS and detached after scrapping in cell lysis buffer (50 mM Tris-HCl, pH 7.6, 0.5 M NaCl, 0.02% NaN_3_, 0.6% NP40, 5 mM EDTA, 1 mM iodoacetamide, 1 mM PMSF, 1 mM Na_3_VO_4_, Protease Inhibitor Cocktail (Sigma)). The Bradford method was used for the protein concentration. Total cell proteins (30 μg) were mixed with 5× Laemmli buffer (Tris 1.25 M, SDS 10%, sucrose 20%, pH 6.8, bromophenol blue 0.005%) and β-mercapto-ethanol, to produce a final concentration of the latter of 3%. The total volume of the samples is maximum 30 μL, with the precondition that all the loading samples have the same concentration. Samples were denatured for 5 min at 95 °C and were then subjected to electrophoresis in a polyacrylamide gel (concentration range: 7.5 to 15%, containing 0.1% SDS). Proteins were transferred by electroblotting onto Hybond-P PVDF membranes (GE Healthcare, Orsay, France), previously activated for 10s in methanol. The membranes were saturated in TBS-T solution (0.1% Tween 20, 20 mM Tris and 140 mM NaCl, pH 7.6) containing 5% nonfat milk (Bio-Rad) or 5% BSA for 2 h at room temperature. Membranes were incubated overnight at 4 °C with constant gentle shaking with primary antibodies (Appendix A). The next day, membranes were washed three times with TBS-T and incubated with a 1:10,000 dilution of the adequate corresponding secondary antibody conjugated to horseradish peroxidase in 1% nonfat milk or 1% BSA, in TBS-T for 1 h at room temperature. After three washes with TBS-T, the bands were revealed by the ECL Prime Chemiluminescence Detection reagent (GE Healthcare, Orsay, France), according to the manufacturer’s instructions. The chemiluminescence signal was captured using a ChemiDocTM MP Imaging (BioRad, Marnes-la-Coquette, France).

### 2.10. Phosphorylation Analysis

Western Blot experiments concerning the signaling pathways require a specific method of preparation of the cell lysates. For every cell line, 5 × 10^6^ cells were required as a starting point. Cells were rinsed twice with a buffer containing 50 mM HEPES, 126 mM NaCl, 5 mM KCl, 1 mM Na_2_EDTA. Then, cells were incubated for 5 min at 37 °C to detach in the above buffer. Serum-free basal medium was added and the cell suspensions were centrifuged for 1 min at 400× *g* in 4 °C. The cell pellets were resuspended in serum-free basal medium and separated in low-binding tubes.

Every tube corresponds to a time point: 0, 5, 10, 15 min. Cell suspensions were prepared both in the absence and presence of lumican (57 kDa). Lumican 100 nM was added at 0 min time point. All samples were placed in a carousel at 37 °C and were pulled according to the time point. All the gathered samples (final volume 100 μL) were placed on ice and then, centrifuged at 400× *g* at 4 °C. Pellets were resuspended with lysis buffer (2% SDS in 50 mM Tris, pH 7.4, 150 mM NaCl, 5 mM EDTA, 10 mM NaF, 2 mM Na_3_VO_4_, 1 mM PMSF, 10 μg/mL leupeptin, 10 μg/mL aprotinin). Samples were centrifuged at 1400× *g* at 4 °C for 15 min. Supernatants were collected and their protein concentration was measured by Bradford Assay. Proteins were denatured at 95 °C for 5 min before western immunoblotting.

### 2.11. Invasion Assay

The invasive potential of B16F1 melanoma cells was evaluated as follows: ThinCertTM cell culture inserts (24-well, pore size 8 μm; Greiner, Bio-One, Courtaboeuf, France) were used and 5 × 10^4^ cells/chamber were seeded in absence or presence of 100 nM of lumican when needed. Every cell invasion chamber includes an 8-micron pore size PET membrane coated with 50 μg of Matrigel. (BD, Biosciences, San Jose, CA, USA). Matrigel. was polymerized at 37 °C for 1 h. In the upper chamber, 200 μL medium used was serum-free with 0.5% BSA, while the lower chamber contains 800 μL of medium, containing 5% FBS, as chemoattractant, and 2% BSA. Negative controls, containing 2% BSA, were also prepared. After 48 h, the non-invasive cells were removed with a cotton swab along with cells that invaded through the Matrigel. The invasion of B16F1 cells was determined by counting the number of stained nuclei by crystal violet under Å~20 magnification using a Zeiss Axiovert-25 inverted microscope equipped with a digital camera (Carl Zeiss). Each individual experiment (n = 3) was made of three inserts per condition and three microscopic fields were counted per insert.

### 2.12. Statistical Analyses

Results were expressed as mean ± SD. Statistical significance between groups was assessed by unpaired Student’s *t*-test or, when necessary, by one-way analysis of variance. *p* value < 0.05 was considered statistically significant. Every standard deviation (SD) was obtained from at least three experiments, made in triplicate. Statistical analysis and graphs were made using GraphPad Prism 6 (GraphPad Software, San Diego, CA, USA).

## 3. Results

### 3.1. Lumican Inhibits Melanoma Metastasis

The effect of lumican on melanoma lung metastasis in wild type and lumican-deleted mice was investigated after 14 days (Figure 1A). Progression of metastasis was promoted by Snail in the absence of lumican (Figure 1A). The number of lung nodules of B16F1 melanoma cells measured as a parameter of metastasis, is shown in Figure 1B. The number of nodules was significantly increased in the case of Snail-B16F1 cells in Lum^−/−^ mice as compared with Lum^+/+^ mice. The total surface of the nodules on lung sections per mouse was measured as a second parameter to evaluate lung metastasis (Figure 1C). A big variability in the size of the tumor nodules in Lum^−/−^ and Lum^+/+^ was observed. No significant differences were observed in the Lum^+/+^ group between Mock and Snail-B16F1 cells in the size of the nodules. Interestingly, the mean surface of the nodules was significantly higher in the Lum^−/−^ group, either injected with Mock-B16F1 or Snail-B16F1 cells.

### 3.2. Evaluation of the Lumican Effect in the Cell Cycle of Melanoma Cells of Lung Metastatic Nodules

Figure 2A shows representative positive staining of cyclin D1 in lung sections (with no visible metastatic nodules) in the Lum^+/+^ mice group (top panel) and lung sections with metastatic nodules of the Lum^−/−^ mice group (bottom panel).

In Figure 2B, the measurement refers to the cyclin D1 positive area (% of cyclin D1 labelled nuclei out of the total number of nuclei per nodules per lung section per mouse) in Lum^+/+^ and in Lum^−/−^ mice groups, both injected either with Mock-B16F1 or Snail-B16F1 cells. The percentage of cyclin D1 positive area was decreased in Snail-B16F1 versus Mock-B16F1 cells in Lum^+/+^ mice group but was similar in Mock and Snail transfected cells in the Lum^−/−^ mice group (Figure 2B). The number of cyclin D1 positive area was higher in the Lum^−/−^ mice group as compared to the Lum^+/+^ mice group (Figure 2B).

In Figure 2C, the number of cyclin D1 positive nodules was similar in Mock and Snail B16F1 cells in the Lum^+/+^ mice group, but it was significantly lower in Snail versus Mock B16F1 cells in the Lum^−/−^ mice group. The number of cyclin D1 positive nodules was higher in the Lum^−/−^ mice group as compared to the Lum^+/+^ mice group (Figure 2C).

### 3.3. Scanning Electron Microscopy Reveals Cells Morphological Changes Induced by Lumican

SEM was performed to investigate the potential lumican regulatory effect on melanoma cells (Mock-B16F1 and Snail-B16F1) showing morphological characteristics related to cancer cell aggressiveness. In particular we evaluated: (a) cell phenotypes, (b) cytoplasmic protrusions like lamellipodia, filopodia and invadopodia, and (c) microvilli, exosomes and microvesicles.

As shown in Figure 3A (images a–c) most of the untreated Mock-B16F1 melanoma cells, cultured for 24 h in serum-free conditions, appeared as isolated, slightly flattened, rounded cells. They did not exhibit evident lamellipodia, filopodia or invadopodia, although the cytoplasmic surface of most cells was covered by many microvesicles, which are known as expression of cell aggressiveness. At higher magnifications, it was possible to observe that some of the same cells assumed a “funnel” shape while they were crossing the Millipore^TM^ membrane-filter pores (Figure 3A, image c). These morphological aspects suggest that untreated Mock-B16F1 cells invade the membrane-filter coated with type I collagen by an invaginating amoeboid movement (“funnel” shape).

When Mock-B16F1 cells were treated with lumican for 24 h, they appeared more globular and still with very few filopodia and no lamellipodia. Most of them assembled in small groups, with cell–cell tight contacts (Figure 3A, image d). No microvesicles were detectable and only few microvilli were present on the cytoplasmic surface, which appeared very smooth. Moreover, in areas where collagen was more abundant and completely filled the pores of the membrane-filter, no invadopodia development was observed (Figure 3A, images d–f). As a general remark, Mock-B16F1 cells upon treatment with lumican showed very few or sometimes no microvesicles on the cell surface, which appeared smoother in comparison with the untreated Mock-B16F1. After 24 h of treatment, lumican seems to reduce the aggressiveness of these cells by favoring their grouping and cell–cell tight contacts. Moreover, lumican modified the method of invasion/invagination of these cells, which did not change their globular original shape while crossing the membrane-filter pores.

When Mock-B16F1 cells were cultured for 48 h they showed a globular shape and were gathered in groups of few and sparse cells which presented no tight cell–cell contacts. Their cell surface was very rough because of the presence of both microvesicles and microvilli (Figure 3A, images g–i). After 48 h of culture with lumican, Mock-B16F1 cells included cells exhibiting cell–cell tight contacts and were more grouped in comparison with the same untreated cells. The cytoplasmic surface displayed rare microvesicles and few microvilli, thus still resulting as smoother vs. the same untreated cells (Figure 3A, images k,l). Some of the globular cells appeared almost completely smooth as they were free of any membrane protrusions (Figure 3A, image l).

SEM observations of Snail-B16F1 melanoma cells cultured for 24 h, confirmed the presence of mesenchymal or morphological elongated phenotypes. They appeared both as isolated or relatively grouped cells and, even though most showed a globular shape, elongated mesenchymal phenotypes were also present. Some grouped cells were involved in a collective invasion process, with the leader cell driving the others through the membrane-filter pores (Figure 3B, image a). The elongated mesenchymal-shaped cells showed lamellipodia, filopodia and invadopodia while penetrating the membrane-filter coated with type I collagen (Figure 3B, image b). Moreover, globular cells displaying microvesicles developed relatively long microvilli which adhered to the membrane-filter coated with type I collagen (Figure 3B, image c).

In contrast, Snail-B16F1 melanoma cells cultured for 24 h in presence of lumican appeared as grouped cells exhibiting cell–cell tight contacts and few microvesicles on their surface, while others appeared almost completely smooth with no microvesicles or microvilli (Figure 3B, images d–f). Thus, after 24 h of treatment, lumican inhibits the mesenchymal phenotypes, which are indicative of the individual invasion by development of invadopodia.

After 48 h, the Snail-B16F1 cells included two different phenotypes: globular cells and mesenchymal-elongated cells. The latter displayed filopodia and lamellipodia, while migrating towards membrane-filter pores in order to invade as individual cells (Figure 3B, image g). However, collective invasion processes were also detectable, with the leader cell developing invadopodia penetrating type I collagen. The following grouped cells were in tight contact to each other and exhibited microvilli on their surface (Figure 3B, images h,i).

After 48 h of culture and treatment with lumican, Snail-B16F1 cells appeared as both isolated and grouped globular cells showing cell-cell contacts and a relatively smooth cell surface with no evident microvesicles (Figure 3B, images j–l). After 48 h of treatment, lumican favors cell grouping, reduces microvesicles development and prevents the individual invasion of Snail-B16F1 cells by inhibiting the development of mesenchymal, elongated cells. Lumican treated cells seem to be still able to cross the membrane-filter pores by an amoeboid movement which did not involve a deformation of the original globular cell shape.

### 3.4. Lumican Alters the Expression and Distribution of Major Invadopodia and Focal Adhesion Markers

The expression of cortactin, marker of invadopodia, and vinculin, marker of focal adhesions, was analyzed by immunofluorescence. Confocal microscopy revealed that cortactin was abundantly expressed both in the cytoplasm and the protrusions of Mock-B16F1 cells. (Figure 4A(a–c)). The picture at higher magnification (3×) and the arrows shows the strong co-localization of cortactin and actin in the focal adhesions (Figure 4A(b)). Treatment of Mock-B16F1 with lumican resulted in a reduced expression of cortactin (Figure 4A(c)). Cortactin was highly expressed in the Snail-B16F1 melanoma cells and its positive expression in the cytoplasmic protrusions is depicted by the arrows (Figure 4A(d,e)). The presence of lumican did not alter in cortactin distribution (Figure 4A(f)). However, as shown in Figure 4A(f), cortactin expression was rather decreased in the focal adhesions. Taking into consideration these data as well as the inhibitory effect of lumican in migration of Snail-transfected melanoma cells it is plausible to suggest that lumican prevents the remodeling of the distribution of major invadopodia markers in Snail-B16F1 melanoma cells [13].

In respect to vinculin, confocal immunofluorescence analysis of Mock-B16F1 melanoma cells revealed an abundant expression of vinculin mostly in the cytoplasm (Figure 4A(g)). The co-localization of vinculin and actin in the cytoplasmic protrusions was observed. Images h,i (Figure 4A) show the co-localization with actin in agreement with the fact that vinculin is an actin-binding-membrane-cytoskeletal protein in cell-ECM junctions. In the presence of lumican, the staining of vinculin was strongly reduced (Figure 4A(j)). In Snail-B16F1 cells (Figure 4A(j)), vinculin expression was decreased in comparison to Mock-B16F1 cells. However, as observed at the higher (3×) magnification (Figure 4A(k)), the localization of vinculin was still present in the cytoplasmic protrusions, although to a lower extent. Arrow in the insert pinpoints vinculin expression in the most distant extension of filopodia (Figure 4A(k)). Snail-transfection had a drastic effect on vinculin expression and distribution. Incubation of Snail-B16F1 cells with lumican (100 nM for 48 h) affected the expression of vinculin, which was not present in filopodia and lamellipodia (Figure 4A(l)). This reduced expression of vinculin in the presence of lumican in the highly metastatic Snail-B16F1 cells underlines the anti-cancer action of lumican.

The expression of cortactin and vinculin was also analyzed by western immunoblotting in the Mock- or Snail-transfected B16F1 melanoma cells. The obtained data are interpreted qualitatively due to the absence of an appropriate loading control. As shown in Figure 4B, cortactin (85 kDa) was highly expressed in the Mock-B16F1 cells and seems to be constant or slightly increased in presence of lumican. In contrast, cortactin expression was reduced in Snail-B16F1 cells as compared to Mock-cells independently of the absence or presence of lumican. Vinculin (124 kDa) follows similar pattern with cortactin in western blotting experiments. Vinculin in the Mock-B16F1 cells treated with lumican seem to be increased as compared to non-treated cells. On the other hand, in Snail-B16F1 melanoma cells treated with lumican, vinculin expression is reduced (Figure 4B). In addition, Snail-transfection decreased the expression of vinculin as compared to Mock B16F1 cells in absence or in presence of lumican.

### 3.5. Lumican Inhibits HA Synthesis, Matrix-Degrading Effectors and Cell Invasion

The effect of lumican on invadopodia markers, prompted us to investigate the effect of lumican on HA synthesis markers, i.e., *HAS2* and *HAS2-AS1*. The gene expression of *HAS2* and *HAS2-AS1* were drastically increased in Snail-B16F1 cells as compared with the Mock ones (Figure 5A). Notably, lumican was able to prevent the Snail-induced gene expression of *HAS2* and *HAS2-AS1* (Figure 5A).

The expression of heparanase and MMP-14 was analyzed by qPCR and western immunoblotting (Figure 5A,B). Heparanase gene expression was increased in Snail-B16F1 melanoma cells as compared to the control Mock-B16F1. Lumican attenuates heparanase expression in Snail-B16F1. It is worth noticing that lumican downregulates the MMP-14 gene expression in both Mock-B16F1 and Snail-B16F1 melanoma cells (Figure 5A,B).

The effect of lumican in cell invasion was evaluated following cell cultures for 48 h on type I collagen coating in the absence and presence of lumican. The Snail-B16F1 cell line proved to be significantly more invasive than the Mock-B16F1 cells (Figure 5C). In the presence of lumican, the number of cells migrating through the pre-coated inserts was significantly decreased for both melanoma Mock- and Snail-B16F1 cells. Specifically, the invasive capacity of Mock-B16F1 cells was inhibited by 25%, and in the case of Snail-B16F1 cells the inhibition reached 40% in the presence of lumican.

### 3.6. Lumican Decreases the Phosphorylation of Proteins Involved in the Regulation of Cell Migration and Cell Growth

To compare whether lumican could act through a similar mechanism in melanoma cells transfected with Snail, FAK phosphorylation status was studied by western immunoblotting in Snail-B16F1 cells (Figure 6A). The level of phosphorylation of FAK at tyrosine 397 was similar when the Mock-B16F1 cells were pre-incubated with lumican for several time points as compared with non-treated cells (data not shown). In the absence of lumican, the analysis of the kinetic of phosphorylation at FAK-pY397 of Snail-B16F1 cells shows that the phosphorylation occurs after 10 to 15 min (Figure 6A). In presence of lumican, after 15 min of incubation of Snail-B16F1 cells, the FAK-pY397/FAK ratio was significantly decreased by 40% as compared with FAK-pY397/FAK ratio in absence of lumican (Figure 6A). Consequently, the phosphorylation of several mediators in cell signaling was investigated with samples obtained after 15 min incubation of cells with 100 nM lumican (Figure 6B). The phosphorylation analysis of the proteins specifically involved in signal transduction was conducted by western immunoblotting. An alteration in phosphorylation of numerous proteins was found. Lumican significantly decreased the phosphorylation of AKT (ser473), ERK 1/2- MAPK 42/44, GSK3α/β, and p130 Cas proteins (Figure 6B).

The expression of total proteins was not altered. These results indicate that lumican was a regulator of many cell signaling pathways.

## 4. Discussion

Previous in vivo study demonstrated that the size of primary melanoma tumors was significantly smaller in the group of wild-type mice (Lum^+/+^) injected subcutaneously with Snail-B16F1 cells in comparison with the mice deficient in lumican gene (Lum^−/−^) [13]. Moreover, the stimulatory effect of SNAIL in the MMP-14 activity and the in vitro melanoma cell migration countered by lumican and its derived peptide in the MMP-14 activity [25,33,34] led us to investigate the effect of lumican in vivo.

The present study demonstrated the pro-invasive role of SNAIL and the inhibitory effect in the formation of pulmonary metastasis by endogenous lumican after intra vascular injection of mock and Snail-B16F1 cells in wild type and lumican deficient mice (Lum^−/−^).

The results show that endogenous lumican inhibits the cell proliferation in lung metastatic nodules of melanoma cells. Progression of cell cycle is regulated by cyclins, cyclin-dependent kinases, and the respective cyclin-dependent kinase inhibitory proteins. Cyclin D1 overexpression is reported to correlate with short cancer patient survival, as well as tumor progression, and can lead to oncogenesis by inducing anchorage-independent growth and angiogenesis through VEGF promotion [36] In the present report, cyclin D1 was abundantly expressed in the lumican knockout-mice, injected with either Mock-B16F1 or with Snail-B16F1 melanoma cells, while it was less expressed in the wild type lumican group, rendering lumican a powerful metastasis suppressor.

Analysis at SEM demonstrated that lumican treatment reduces the aggressiveness of both Mock-B16F1 and Snail-B16F1 cells by reducing the cytoplasmic protrusions and microvilli which are related to cancer cells’ aggressiveness [24]. Lumican treatment also favors grouping of cells and cell–cell tight contacts and modifies the method of cell invasion/invagination in Mock-B16F1. It is of interest that in Snail-B16F1 cells, lumican also inhibits the mesenchymal phenotypes which are expression of individual cell invasion through the invadopodia development and a proteolytic process [24,37].

Invadopodia formation was analyzed first by confocal microscopy and later by immunostaining following the distribution and expression of two specific biomarkers. Cortactin is a regulator protein of the formation and structure of invadopodia [38]. The co-localization with actin is expected, since cortactin is essential for the maintenance of F-actin-enriched invadopodia core structures. The direct relation of SNAIL and the number of formed invadopodia and particularly the expression of cortactin, marker of invadopodia, has been shown. Lumican significantly inhibited the expression of cortactin. Inhibition of cortactin leads to the blocking of Src signaling, which results in the inhibition of invadopodia formation and, in the long term, the inhibition of metastasis, confirming the anti-metastatic action of lumican [39].

Vinculin overexpression facilitates cell adhesion and recruitment of cytoskeletal proteins at the domains of integrin binding at the sites of focal adhesions [40]. Inhibition of vinculin leads to the alteration of many cell functions, such as formation of fewer focal adhesions and inhibition of lamellipodia protrusions [41]. Vinculin is part of the focal adhesion complexes and consists of essential invadopodia markers [42]. Thus, the role of vinculin in the morphology of the melanoma cells was addressed. Regarding vinculin staining, its expression was significantly decreased by the treatment of lumican at the focal adhesion complexes, accompanied by decreased formation of invadopodia as observed by confocal microscopy. Vinculin increased expression facilitates cell adhesion and recruitment of cytoskeletal proteins at the domains of integrin binding at the sites of focal adhesions. On the other hand, in Snail-B16F1 melanoma cells treated with lumican, vinculin expression is significantly reduced, leading to the formation of fewer focal adhesions and inhibition of lamellipodia protrusions, further suggesting the anti-metastatic effect of lumican.

The effect of lumican on invadopodia markers, prompted us to monitor the effect of lumican of hyaluronan synthesis markers, such as synthase HAS2 and HAS2-AS1. Hyaluronan synthesis is induced not only by the cross-talk between stromal and tumor cells, but also by tumor cells themselves promoting their aggressive potential [43]. Moreover, antisense inhibition of HAS2 in aggressive breast cancer has been reported to inhibit the in vivo formation of tumors [17]. It is known that the metastatic ability of cancer cells is associated with the degradation of the ECM and increased heparanase expression. Heparanase degrades heparan sulfate chains and it contributes to ECM remodeling. Heparanase is eventually involved in ECM remodeling and EMT [44,45]. When heparanase is downregulated, it can inhibit tumor metastasis [44,45]. Previous studies from our laboratories showed that MMP-14 activity was inhibited by lumican in melanoma [33].

Cell invasion is a crucial biological property, and therefore the effect of lumican on melanoma cell invasion properties was analyzed by an in vitro cell invasion assay (Figure 5C). We investigated whether lumican could prevent invasion through type I collagen coating. The demonstrated inhibitory effect of lumican on melanoma cell invasion may suggest that lumican potentially inhibited cell migration by involvement of FAK phosphorylation in EMT-like melanoma cells.

MMP-14 plays an important role in cell migration, not only by regulating the activity and expression of downstream MMPs [46], but also by processing and activating migration-associated molecules, such as integrins or CD44. MMP-14 interacts with CD44 and localizes at invadopodia, rendering MMP-14 a crucial regulator of cell migration and invasion. Lumican was shown to be an inhibitor of MMP-14 expression and thus of cell invasion and metastasis.

MMP-14 also activates a variety of intracellular signaling pathways, such as MAPK, FAK, Src and Rac [47,48]. The inhibition of melanoma cell migration by lumican was related to the inhibition of the phosphorylation of focal adhesion kinase (FAK) [28], and as observed, a significant decrease of the ratio pFAK/FAK was also shown in presence of lumican [28]. In this study, in the presence of lumican, inhibition of pY397FAK phosphorylation was also observed. The ratio pFAK/FAK was decreased by 40% compared with control. Therefore, the decreased FAK phosphorylation and the inhibition of MMP-14 activity induced by lumican in melanoma cells might explain, at least in part, the anti-invasive effect of this SLRP.

Proteolytic degradation of ECM by MMP-14 induces integrin cell–cell and cell–matrix contacts, focal adhesion stability, FAK phosphorylation at Tyr-397, AKT activation, and ERK activation, resulting in the promotion of cell migration [47]. Our hypothesis is that lumican inhibits the degradation of ECM by inhibiting MMP-14, then influencing integrin clustering, modulating focal adhesion stability and FAK phosphorylation at Tyr-397, and inhibiting cell migration. Furthermore, it was also demonstrated that activation of RTK, like EGFR, leads to a phosphorylation of FAK and promotes cell migration [48]. Therefore, lumican’s anti-cancer effect in respect to the decrease of EGFR phosphorylation and the downregulation of the downstream signaling, which results in decreased phosphorylation of FAK and inhibition of cell migration, could be considered as a potential mechanism that needs further evaluation.

The inhibition of FAK phosphorylation induced by lumican leads to downregulation of the phosphorylation of p130 Cas and AKT. The inhibition of p130 Cas phosphorylation makes this protein inactive. As a consequence, a down-regulation of the downstream signaling events may explain the inhibition in lamellipodia formation and MMP production resulting in an inhibition of cell migration. Moreover, the inhibition of AKT kinase results in a decrease of its activity and to a reduction of the downstream signaling cascade which may cause inhibition of cell growth. The AKT–mediated phosphorylation of serine-9 in GSK3α/β significantly decreases the availability of the active site for its substrate, such as β-catenin. However, lumican by inhibiting phosphorylation of AKT causes decreased phosphorylation of GSK3α/β at serine-9 resulting in possible increased activity of this enzyme. Degraded proteins cannot be translocated to the nucleus and act as transcription factors for genes connected with cell proliferation and cell migration. This may explain decreased melanoma cell growth and cell motility in the presence of lumican. Further investigation to evaluate the involvement of MAPK pathway related to MMPs as well as cyclin D1 transcription and p21, which is a cyclin dependent kinase inhibitor, will be useful to evaluate the underlying mechanisms of the lumican action and its potential use for future pharmacological disease targeting.

Altogether, the results are summarized in Figure 7. These results are promising and indicate a potential mechanism of action for lumican in melanoma metastasis inhibition. Functionalized nanoparticles containing or coated with lumican (or lumican-derived peptides able to mimic the anti-tumor effect of lumican) could be inoculated in vivo by intravascular injection in mice melanoma models to investigate whether it could down-regulate the development of lung metastatic nodules of B16F1 melanoma cells. Then, if the effect is demonstrated in animal models in pre-clinical assays, and that no cytotoxic effect is demonstrated, potential clinical assays could be designed.

## Figures and Tables

**Figure 1 cells-10-00841-f001:**
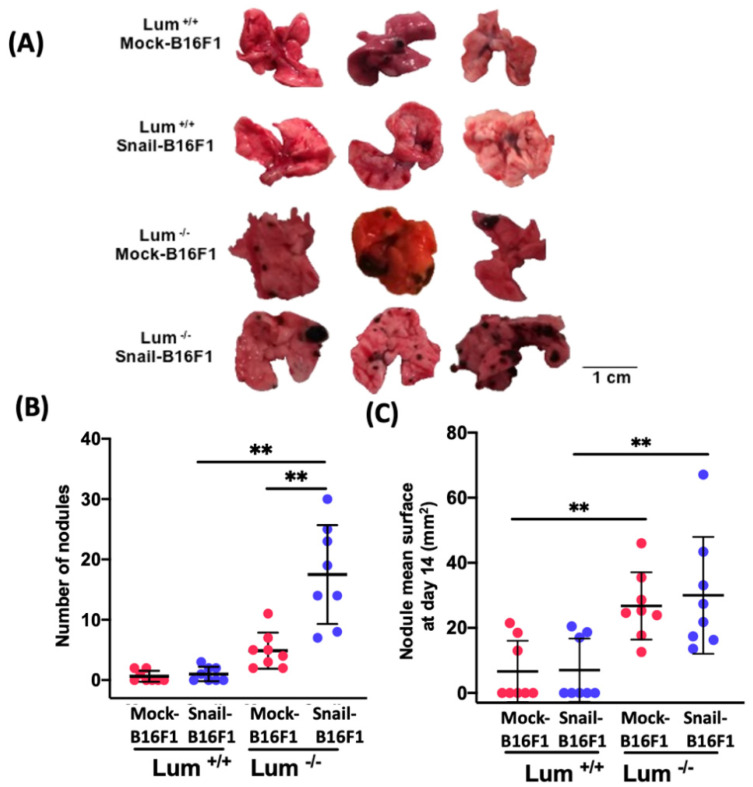
Study of lung metastatic nodules of melanoma cells in the wild-type and lumican-knockout C57BL/6J mice. (**A**) Lung metastatic nodules, observed at day 14, obtained after injection in the tail vein (n = 250.000 cells/mouse) of Mock-B16F1 and Snail-B16F1 (three representative pictures are shown from each group (n = 8) of Lum^−/−^ and Lum^+/+^). Lung metastatic nodules are significantly more numerous in the Lum^−/−^ in comparison with the Lum^+/+^ mice. (**B**) Number of metastatic nodules detected per mice at day 14. (**C**) Total mean surface of the nodules measured per section of lung tissue per mouse at day 14. Statistical differences at the level of *p* < 0.01 are given with asterisks (**).

**Figure 2 cells-10-00841-f002:**
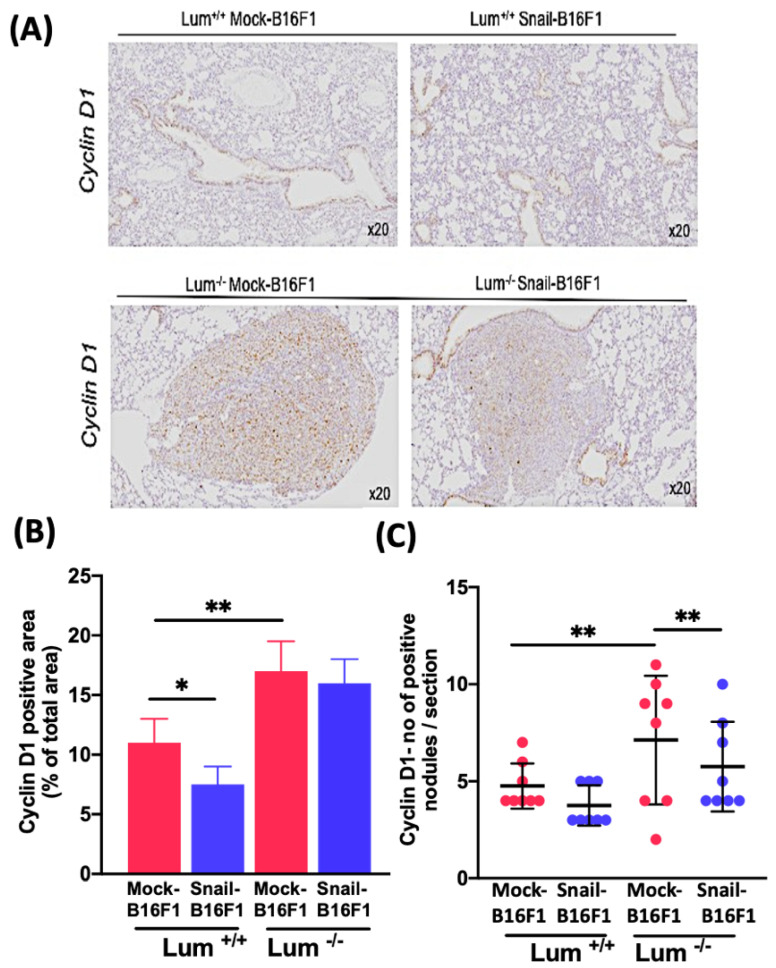
Evaluation of the expression of cyclin D1 in the Lum^+/+^ and Lum^−/−^ C57BL/6J mice. (**A**) Immunohistochemistry of cyclin D1 in mice lungs of Lum^+/+^ and Lum^−/−^ mice injected either with Mock-B16F1 or Snail-B16F1 melanoma cells. The immunostaining was localized to the cell nuclei. Hematoxylin counterstain; original magnification 20×. (**B**) Measurement of the cyclin D1 positive area (Cyclin D1 positive nuclei normalized to the total number of nuclei in lung nodules per lung section per mouse) in the Lum^+/+^ and Lum^−/−^ group of mice, injected either with Mock-B16F1 or Snail-B16F1 cells. Cyclin D1 expression was found positive in both groups. (**C**) Measurement of the number of cyclin D1 positive nodules per section. Graphs represent the mean value ± standard deviation (n = eight mice per group, each symbol represents the value for every mouse) Statistical differences at the level of *p* < 0.05 and 0.01 are given with asterisks (*) and (**), respectively.

**Figure 3 cells-10-00841-f003:**
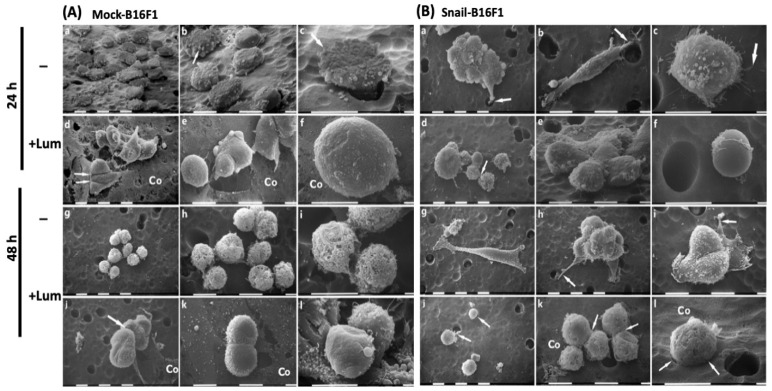
SEM of Mock-B16F1 and Snail-B16F1 melanoma cells melanoma cells cultured in serum-free conditions on a Millipore^TM^ membrane-filter and coated with type I collagen before and after treatment with lumican for 24 and 48 h. (**A**) Untreated Mock-B16F1 melanoma cells, cultured for 24 h (images (**a**–**c**)). A “funnel” shaped cell with microvesicles (arrow) and crossing a Millipore pore is shown in image **c**. Cells treated with lumican (100 nM) are shown in images (**d**–**f**). They assemble in small groups, with tight cell–cell contacts (arrows) (Co indicates the collagen coating). Untreated Mock-B16F1 cells cultured for 48 h are given in images (**g**–**i**). Following treatment with lumican for 48 h, Mock-B16F1 cells exhibiting tight cell–cell contacts and appear more grouped (images (**j**–**l**)). The globular cell on the left side appears almost completely smooth, free of any membrane protrusions (image (**l**)). (**B**) Untreated Snail-B16F1 melanoma cells cultured for 24 h (images (**a**–**c**)). Cells are involved in collective invasion with the leader cell developing an invadopodia (arrow) (image (**a**)). A mesenchymal or morphological elongated cell invades following an individual invasion process (image (**b**)). A globular cell displaying microvesicles shows relatively long microvilli (arrow), which anchors to type I collagen coating (image (**c**)). Snail-B16F1 melanoma cells treated with lumican for 24 h are shown in images (**d**,**e**). A single globular cell shows a completely smooth surface with no cytoplasmic protrusions like microvilli or microvesicles (image (**f**)). Untreated Snail-B16F1 cells are shown in images (**g**–**i**). A mesenchymal-elongated cell showing lamellipodia is visible in image g. In the collective invasion process, the leader cell developing invadopodia (arrow) and the following grouped cells are in tight contact to each other and exhibit microvilli on their surface (images (**h**,**i**)). Snail-B16F1 cells treated for 48 h with lumican are shown in images (**j**–**l**). They appeared as both isolated (arrows) (image (**j**)) and grouped globular cells showing both cell-cell contacts (arrows) and relatively smooth cytoplasmic surfaces with no evident microvesicles (image (**k**)). A single globular cell is penetrating a membrane-filter pore (arrows) by an amoeboid movement with no deformation of the original globular shape (images (**l**)). All images were taken from the top of the Millipore filter so that cells are passing through or preparing to cross the barrier. White bars = 10 µm. Three different samples were used for every experiment.

**Figure 4 cells-10-00841-f004:**
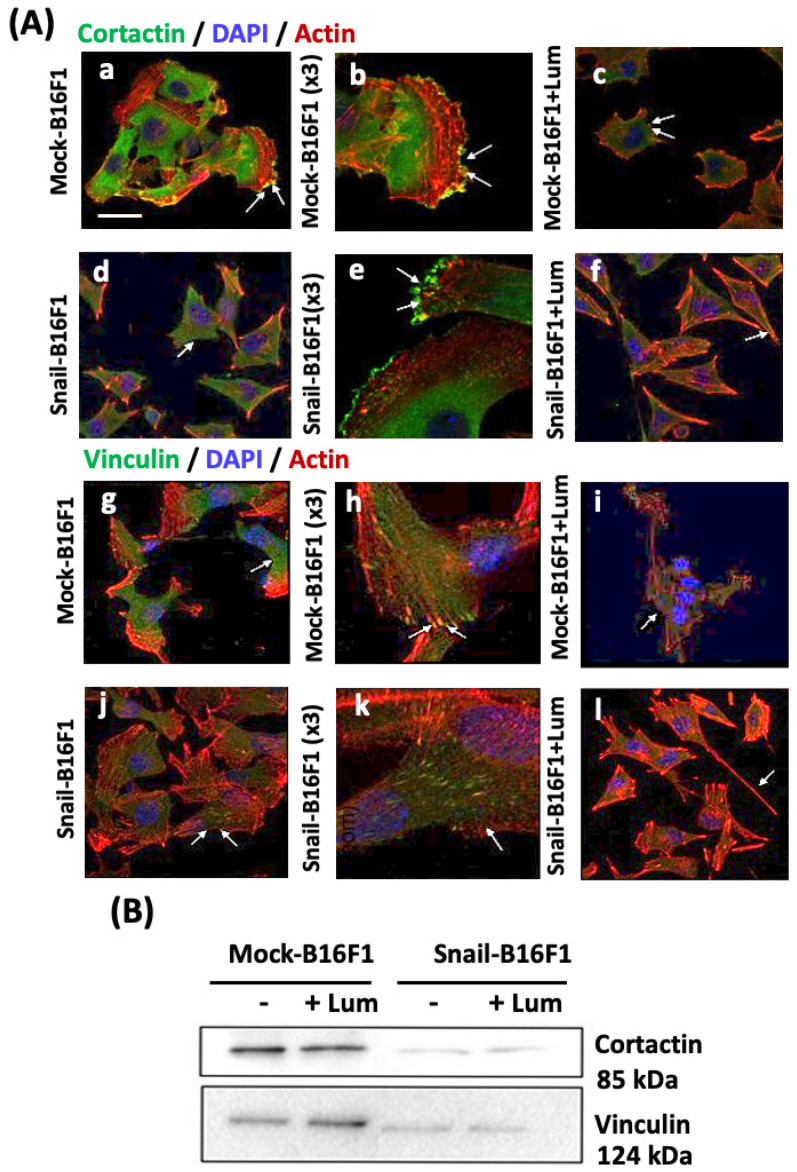
Distribution and expression of cortactin and vinculin in Mock-B16F1 and Snail-B16F1 melanoma cells studied by confocal immunofluorescence and western immunoblotting. (**A**) Cortactin (top panel, (**a**–**f**)) was detected all along the membrane protrusions in Mock-B16F1 melanoma cells (**a**–**c**). Arrows show the strong co-localization of cortactin and actin in the focal adhesions. Lumican treatment (**c**) resulted to lower expression of cortactin. Cortactin was highly expressed in the Snail-B16F1 melanoma cells (**d**–**f**) and its positive expression is depicted by the arrows. In the presence of lumican, cortactin staining reduced in the protrusions (**f**). Vinculin (bottom panel, (**g**–**l**)) was abundantly expressed in the Mock-B16F1 cells (**h**,**i**). Vinculin is located in the rings of the periphery of newly created invadopodia co-localized with actin in the cytoplasmic protrusions. In Mock-B16F1 treated with lumican, vinculin was still present in the cytoplasm (**i**), but reduced as compared with the control (**g**,**h**). In Snail-B16F1 (**j**–**l**), vinculin immunolabelling can still be detected (arrows), but was weaker in the presence of lumican (**l**). The arrows in the insert pinpoint vinculin expression in the most distant extension of filopodia. Scale bar: 10 μm for all images, except the zoom ones (×3). (**B**) Western immunoblotting of cortactin and vinculin in Mock-B16F1 and Snail-B16F1 cells before and after treatment with lumican (100 nM) for 48 h.

**Figure 5 cells-10-00841-f005:**
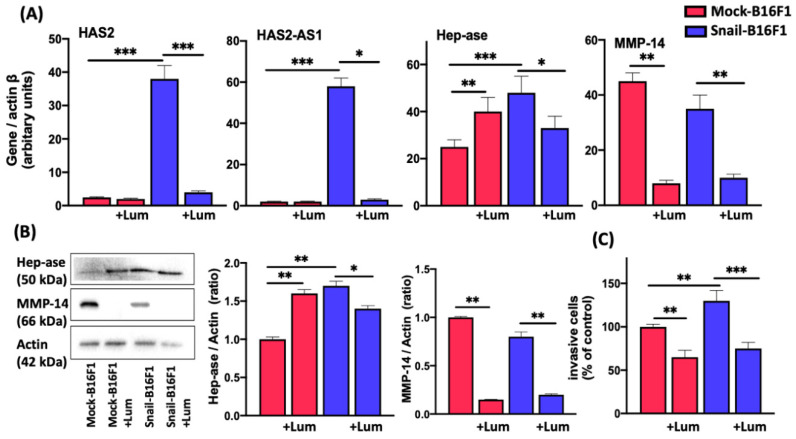
Lumican’s effects on hyaluronan biosynthesis players, matrix-degrading components and cell invasion. (**A**) qPCR analysis of the HA biosynthesis markers HAS-2 and HAS2-AS1 and of Heparanase and MMP-14. HAS2 gene expression is very low in Mock-B16F1 in absence or in presence of lumican. In contrast, HAS2 gene expression is high in Snail-B16F1 cells. Lumican significantly attenuates the gene expression of HAS-2 (40 times) in Snail-B16F1 melanoma cells. The HAS2-AS1 gene expression is very low in Mock-B16F1 in the absence or presence of lumican. In contrast, HAS2-AS1 gene expression is high in Snail-B16F1 cells. HAS2-AS1 was drastically decreased in Snail-B16F1 cells (from 58 to 0.5) in presence of lumican. Heparanase gene expression in Mock-B16F1 is significantly stimulated in presence of lumican. In addition, Heparanase gene expression is high in Snail-B16F1 cells as compared to Mock-B16F1 cells but is decreased by lumican treatment. MMP-14 gene expression is high in Mock-B16F1 and in Snail-B16F1 cells but it is significantly decreased in presence of lumican in both cell types. (**B**) Protein expression of heparanase and MMP-14 in Mock-B16F1 and Snail-B16F1 melanoma cells incubated 48 h with lumican (100 nM) analyzed by western immunoblotting and normalized to actin. The protein expression of heparanase and MMP-14 varies in a similar manner to that of their gene expression. (**C**) The effect of lumican in cell invasion was evaluated following cell cultures for 48 h in the absence and presence of lumican using type I collagen as substrate. Untreated Mock-B16F1 cells (control) exhibited significantly lower invasion potential than Snail-B16F1 cells. Lumican significantly decreased the invasion of both cell lines, with more profound effect in Snail-B16F1 cells. Each bar represents the mean of three independent experiments analyzed in triplicate. Asterisks indicate statistically significant differences (*, *p* < 0.05; **, *p* < 0.01; ***, *p* < 0.001).

**Figure 6 cells-10-00841-f006:**
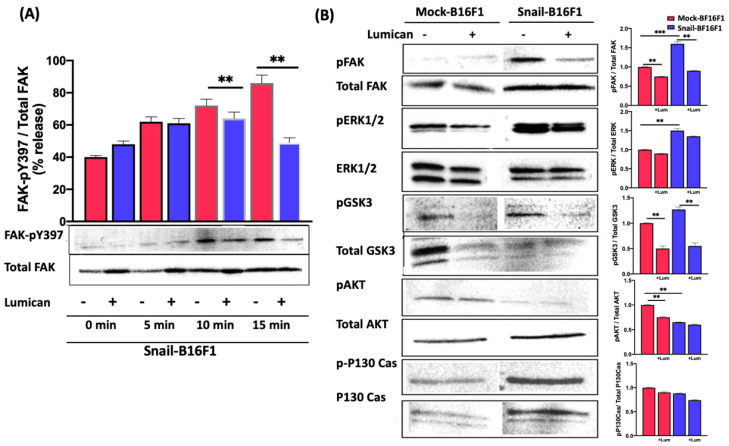
Lumican decreases the phosphorylation of specific kinases involved in signal transduction. (**A**) The kinetic of Phospho-FAK (pY397) expression and total FAK expression in Snail-B16F1 cells before (red bars) and after incubation (blue bars) with lumican (100 nM) for 0, 5, 10, 15 min was analyzed by western immunoblotting. After densitometric analysis of the intensity of the bands, the resulting ratios of FAK-pY397 to total FAK intensity were calculated and depicted graphically. Data are presented as mean values ± standard deviation from three independent experiments, (**B**) The phosphorylation of specific proteins involved in signal transduction in Mock-B16F1 and Snail-B16F1 cells with lumican (treatment with 100 nM for 15 min) was analyzed by western immunoblotting. Lumican decreased the phosphorylation of specific intracellular signaling mediators, involved in signal transduction. After densitometric analysis of the intensity of the bands, the resulting of phosphorylated mediators to total protein intensity was calculated and depicted graphically. Three samples were analyzed (**, *p* < 0.01; ***, *p* < 0.001).

**Figure 7 cells-10-00841-f007:**
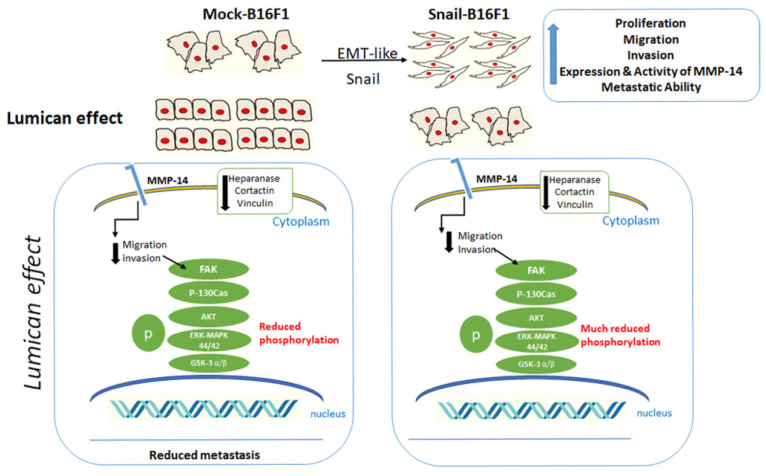
Diagram summarizing the major effects of lumican in melanoma cells before and after Snail transfection and potential lung metastasis. The action of lumican has been classified according to its effect on cell morphology, invadopodia markers (cortactin, heparanases), and RTKs (FAK, AKT, ERK-MAPK 44/42) according to Snail overexpression.

## Data Availability

The study does not report any data.

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
