# Peer review of "Lumican Inhibits In Vivo Melanoma Metastasis by Altering Matrix-Effectors and Invadopodia Markers"

_cells, 2021, doi:10.3390/cells10040841_

Round 1

Reviewer 1 Report

Summary: In the revised version, Karamanou et al. have responded to many of our points. However, many issues remain, and furthermore with their extensive changes, new issues have surfaced. The descriptions of figures still are not complete or clear, and several concerns with data and design have not been addressed.

Highlighted numbers refer to the original points in the first set of reviewer comments.

Major points:

  • The manuscript in its present form does not appear to be completed and formatted for final submission. The extensive tracked changes make it very difficult to read the ms in a continuous way, as the reader has to search through lines of text to find the continuation of sentences. Text added in this version should be highlighted. (Note: the clean copy sent separately by the editor greatly assisted with readability, but neither it nor the original copy has changes marked with highlighting as the authors state in response to reviewers.)

  • The introduction is now more comprehensive, and has improved in some ways, but the topics explained do not follow a logical order. In other words, it is lacking organization. There is not a good flow from paragraph to paragraph. For instance, why was this order chosen? The topics jump from cyclins to MMPs to EVs to ECM factors to EMT and then back to metastasis.
    The relevance of the study is better explained with the additional general statistics; however, it is still unclear whether metastatic melanoma is a common problem. What percent of patients are diagnosed with late-stage disease and end up with metastasis?

  • The text in response to reviewers (answer #1d) does not appear in the manuscript:

“Cancer cell-derived extracellular vesicles (EVs) are increasingly being recognized as genuine invasive structures as they contribute to many aspects of invasion and metastasis. Recent reports indicate a role of the actin cytoskeleton in the mechanisms underlying EV biogenesis or release. Indeed, Els Beghein and collaborators have demonstrated a role of the cortactin in EV release. A contribution of this protein in endosomal trafficking was shown to be a crucial step in EV biogenesis. EVs are preferentially released at invadopodia, the latter being actinrich invasive cell protrusions in which cortactin performs essential roles. Accordingly, EVs are enriched with invadopodial proteins such as the matrix metalloproteinase MT1-MMP and exert gelatinolytic activity. Thus, cortactin plays key roles in EV release by regulating endosomal trafficking or invadopodia formation and function.”

  • Figure 1: In either the figure or legend of 1A, it would be preferable to add number of mice (in addition to having this info in the methods). For panels B and C, the groupings are confusing with the brackets. Make x-axis grouping titles (Mock-B16F1, Snail B16F1…) perpendicular to the x-axis and only add straight line to indicate grouping of Lum+/+ vs Lum-/-, not a bracket.

  • Figure 2: the title of this results section is: “evaluation of the lumican effect in the cell cycle of melanoma cells…” The figure is only showing the effect of having no lumican. Line 299 (line numbers refer to the “clean” manuscript): “not shown” is not acceptable; please either include data in Fig. 2 or in supplemental data. The point of this figure is not clear, since there is no difference between mock and Snail-B16F1, and no comparison is made to Lum+/+ (WT). Is the purpose of figure 2 to say that cell cycle progression (due to cyclin D1) is similar between the Mock- and Snail B16F1 Lum-/- mice? In 2C, what do the triangles represent and why do the groups have different numbers? Weren’t 8 mice injected per group?

  • Figures should be presented in final form, and with all descriptive data. For example, Fig. 3 is presented as individual panels, each with its own legend. This is not standard and does not allow for an adequate overview of the proposed figure. The panels are labeled in a haphazard way, with different text sizes and insufficient descriptions of data. Figure legends do not state whether lumican is present or not; there are references to lumican treated cells (24 and 48h) and untreated cells, but it is not clear which images these refer to. It is not clear what belongs to figure legend and what belongs to main text. In Fig. 3A, “Co” indicates collagen on d-f, j,k. Do other panels not have collagen? Scale bars are difficult to interpret, as in many panels bars of different sizes appear.

  • Figure 3: Label the images. First row is the untreated Mock-B16F1 cells, second row is the Mock B16F1 cells treated with lumican, and so on… On image c, how are you able to see a Millipore hole if the membrane has been coated with type I collagen? Clarify whether the images were taken from the top (where cells were plated) or bottom, after invasion. If the images were taken from the top, cells may seem like they are preparing for invasion, but it does not necessarily mean that they did actually invade. Furthermore, if the pores are visible, it could very well mean that the cells caused the degradation of collagen; however, it could also mean that the filters were not properly coated. In 3C, vinculin staining is really hard to see. Is there any way to improve this? Even higher magnification, perhaps? Furthermore, if cells are not necessarily degrading the collagen, passage through the micropores would signify migration, not invasion. “Before” pictures (of the membrane coated with collagen) would be a good control. In panel 3D, if “cortactin is essential for the maintenance of F-actin-enriched invadopodia”, is actin a proper control for Western blot? Would not a protein unrelated to cytoskeletal structure be more appropriate as a control?
    Thank you for your response to point #11. However, the authors did not respond to all issues. Fig. 3C: Is ‘e’ a zoomed portion of d? It is not clear that these are depicting the same cells. Please provide a box on one image to indicate the portion of that image that is zoomed on a subsequent image (e.g. 3Ca,b). Further, the text states that the arrow in d indicates high expression of cortactin, but no green labeling is evident. Because very little green labeling is apparent in f, the author’s statement that distribution of cortactin with lumican treatment is unaltered (line 454) cannot be evaluated. In summary, the quality of the confocal images is not sufficient to warrant the authors’ conclusions.

  • 3: Was the level of Snail expression (as well as other EMT factors) checked after lumican treatment? How common is lower expression of lumican in metastatic cancer patients? Is it a commonly mutated/deleted/dysregulated gene? If it is significantly elevated in the tumor stroma of melanoma (line 98), why is melanoma invasion/metastasis a problem?

  • Please provide consistent scale bars: e.g. Fig. 3C has a scale bar in 1 of the 14 images. Response to reviewers states that scale bars have been added (3Cg, 3Cm, 3Cn), but they are still missing. Every image should have a scale bar.

  • 4: A: No gray bars show up, therefore it is difficult to see where the mock samples are. B: For Western blots, the image for the actin control is the same in figure 3 as in figure 4. It is not acceptable to have duplicated images. Were all proteins stained for in the same membrane? Please provide whole blots. From the visual appearance of the blot, it looks like the heparanase in the Snail +Lum condition would be more than that in the Snail –Lum condition, yet the quantitation shows that it is less. In regard to panel c, what does “absence and presence of lumican” mean: Were cells starved with lumican or was lumican added after starvation? Please add details in methods section. Also, the y-axis is confusing. What does percent of control mean: Is this in relation to cells without serum? What does 150% invasive cells mean? Does it mean cells invaded and then multiplied?

  • Figure 5: In some cases, boundaries between lanes are unclear. When bands are fused together, how is densitometric analysis performed? How is the beginning and end of a band determined?

  • The connection between morphology, actin/cortactin distribution and EMT has not been convincingly presented. Please provide evidence (reference or data) to support the statement (line 456-458) that “… lumican inhibits EMT like of Snail-transfected melanoma cells by preventing the remodeling of the distribution of major invadopodia markers in Snail-B16F1 melanoma cells.” The reviewer does not see evidence presented that an EMT-like event is happening.

  • The micrographs do not support the statement that there is “abundant expression of vinculin mostly in the cytoplasm” (line 459,460). Since the amount of visible green labeling is minimal, the statement that the vinculin expression was strongly reduced with lumican (line 463) cannot be evaluated. Indeed, in the +Lum image provided, actin labeling is also nearly absent. Similarly, because the vinculin labeling is low in the mock condition, the claim that it is decreased in SNAIL overexpressing cells is not supported. The vinculin data should be presented in such a way as to make them observable for the reader.

  • Figure 4: For Western blots, the image for the actin control is the same in figure 3 as in figure 4. Were all proteins stained for in the same membrane? Would it be possible to see the whole blot (at least in supplementary data)? How come the ratio of heparanase/actin is lower in the Snail-16F1 lumican-treated group than the ratio seen in the Snail-B16F1 group without lumican? In regard to panel c, what does “absence and presence of lumican” mean? Were cells starved with lumican or was lumican added after starvation? Please add details in methods section. Also, the y-axis is confusing. What does percent of control mean? Is this in relation to cells without serum? What does 150% invasive cells mean? Does it mean cells invaded and then multiplied?

  • To analyze the WB, having a loading control is necessary. Actin does not appear to perform this role, since its expression varies widely between samples. Also in confocal images, amounts of actin vary widely.

  • Where is the illustration with all structures analyzed at SEM as indicated in response to #14?

  • Please indicate what gray and black bars on Fig. 3D indicate in the figure legend (see comment #15)

  • Please include numbers of samples (biological and technical) in figure legends (see comment #15,16).

  • Please include information from #18 in discussion as requested.

  • Revise English grammar and syntax. There are a lot of simple mistakes and the sentences do not flow easily. Decide whether words really need to be capitalized, like cyclins (lines 60-67), extracellular matrix (line 71), epithelial-to-mesenchymal transition (lines 91-92), Proteoglycans (98), lumican (95), and others.

Minor points:

  • Lines 25-26 consider changing the wording in “invading elongated phenotypes which are known to develop invadopodia”.

  • Lines 31-32: Snail should be one of the keywords since the subject of study is based on Snail-induced EMT.

  • Statements on lines 49-54 require references.

  • Line 147: Check reverse primer sequence. Should the last four nucleotides be TTGA-3’?

  • Line 149: Should the first length be 380-bp or 385-bp? And the second one 300-bp or 301-bp?

  • Line 279: Please remove “and especially”; see minor point #7.

  • Line 292: number of metastatic nodules detected per mouse? (not per group of mice?)

  • Line 313: Immunohistochemistry of cyclin D1 in Lum-/- mouse lungs

  • Line 394, 395: …obliterating the membrane pores (instead of filter holes)?

  • Please add headings to Table S2.

  • A few examples of grammatical / usage errors:
    1. Lines 35-37: ‘Incidences of melanoma are is increasing annually in at a high rate.’
    2. Line 38: life-threatening
    3. Line 39-41: ‘While the environmental factors, the daily lifestyle, as well as and the phenotypic and genetic susceptibility contribute to the initial initiation of melanoma, Melanoma etiology of this disease is complex, multifactorial and heterogeneous.’
    4. Line 42: …displaying a mutagenic role…
    5. Line 45: …nevi play a major role…
    6. Line 52-54: ‘“…human tumorigeneses and progression of metastasis. Progression of metastasis, affecting the development…”
    7. Line 58-59: “cancer cells have been reported to express high levels of MMP-14…” Also, the abbreviation EVs has not been introduced yet. It is first mentioned in Line 60.
    8. Line 60: Cancer cell-derived extracellular vesicles (EVs) are playing play major roles…
    9. Line 61,62: “EVs are mostly released at invadopodia, which are highly rich, invasive cell protrusions, where cortactin plays a regulatory role.”
    10. Line 64-66: This sentence should be rewritten.
    11. Line 66-69: Break this complex sentence into smaller sentences.
    12. Line 69-71. Is the word “it” missing between ‘while’ and ‘decreased’?
    13. Line 73: …proteoglycans, plays a crucial role in the remodeling of…
    14. Line 72-75: consider rewording; separate normal biological function from function in cancer.
    15. Line 78: transcription factor
    16. Line 79: delete situation
    17. Line 82: the meaning of ‘EMT-like’ is unclear; resulted instead of results
    18. Line 96, 97: consider rewording
    19. Line 98, 99: consider rewording
    20. Line 100: remove comma after tissues
    21. Line 100-102: consider rewording
    22. Line 115: as a melanoma model
    23. Line 122: use “in which” instead of “where”
    24. Line 125: …tumor development, while Snail overexpression was shown to induce EMT-like and the metastatic…
    25. Line 139: Extraction of collagen type I was not described in reference 22. Add catalog number of (rh)MMP-14.
    26. Line 282: total surface area of the nodules…

Reviewer 2 Report

The manuscript of Karamanou and coworker is the extension of a series of publications already dealing with the protective role of lumican against melanoma development.  This particular manuscript provide evidence, that the protective role of this proteoglycan works not only in vitro, but in vivo, as well.  Lumican wild type, and lumican -/- C57 black mice were inoculated intravenously with melanoma cells, and their colonization into the lung were followed.  Certainly lumican presence provided prevention against the lung colonization either in mock transfected or Snail transfected tumor cells, the latter indicated that the protection works against EMT transformed cells, as well.

Considering the deadly nature of melanomas, this is a very important information hopefully with future therapeutic development.

In conclusion the manuscript has to be suggested for publication.

In the meantime, before that several issues concerning the presentation of the manuscript has to address.

  1. The English language of the text has to seriously edited as sometimes it is hardly understood, or easily misunderstood. etc: row 55 better: invasion of the tumor to the nearest tissue
  2. There are erroneous statements, such as cyclin D1 and D3 are the major player in melanoma, but although it is an important endpoint, nothing is mentioned about the major injuries in human melanomas, namely mutation of NRas, or BRaf, subsequently MEK, thus the MAPK pathway or sometimes  beta catenin pathway.
  3. row 659: cyclin D1 transcription is promoted by MAPK, or c-myc, but not by beta catenin pathway. Here active GSK3beta put on inactivating phosphorylation on the protein initiating its degradation. p21 is a cyclin dependent kinase inhibitor (and not kinase) transcribed by p53 or Foxo1 (unless she means p21 Ras) ,MMPs (like MMP1 is also transcribed by MAPk route (or NFkB or AKT) and not by the b catenin pathway.
  4. row 65: hyaluronan -the statement of metabolism is OK but should be explained what is this about: glucose metabolism, UDP-glucose dehydrogenase upregulation -HAS2 upr gulation –hyaluronic acid overexpression  –EMT doi: 10.1038/s41388-019-0885-4.
  5. row 72: heparanase must be refer to its involvement in EMT
  6. row 98: the role of lumican, and its expression in young and old cartilage has nothing to do with the story
  7. row 128: here a few sentence are needed about the previous results with lumican, and describe what are the aims in this paper.
  8. row 301: cyclin D1 control sample must be shown
  9. fig2 C I do not understand are there more nodules in the mock, triangles are individual animals? It should  explain .
  10. fig 3D: vinculin fluorescent pictures neither green nor blue is visible, the quality for presentation is better?
  11. Fig4C: Quantitation of MMP14 is wrong. No MMP14 is detected on the Western at snail,B16 lumican, and the graph shows a considerable high column
  12. Fig5: Total ERK- instead EPK, Mock B16 + lumican, the band is stronger than that of Mock B16 alone, grafic evaluation is the opposit.
  13. row 559: again control cyclin D1 is needed (and western?)
  14. row 599 reference for heparanase and EMT is needed doi: 18632/oncotarget.26042
  15. row 563: protrusions like microvesicles micrivesicles are not protrusions
  16. row 569: I would use immunostaining inseas of immunoblotting
  17. row 574-579: the same is described 2 times
  18. row 621: which activating phosphorylation of Akt was studied?

 row 626:  Without measuring I would not say, as final conclusion, that the real mechanism of  lumican action is the inhibition of EGFR activation concluding to the dowregulation of FAK activity. Needless to say it can considered as a potential mechanism,  to study in the future.

Author Response

we thank this reviewer for recognizing the importance of the current
research and the constructive comments made. We have taken all comments into consideration in the revised version which was corrected accordingly. Changes are made with “track changes” and also highlighted in yellow. The comments raised held us to improve the quality of the manuscript.
Below a point-by-point response to reviewer’s comments is given.

The manuscript of Karamanou and coworker is the extension of a series of publications already dealing with the protective role of lumican against melanoma development. This particular manuscript provide evidence, that the protective role of this proteoglycan works not only in vitro, but in vivo, as well. Lumican wild type, and lumican -/- C57 black mice were inoculated intravenously with melanoma cells, and their colonization into the lung were followed. Certainly, lumican presence provided prevention against the lung colonization either in mock transfected or Snail transfected tumor cells, the latter indicated that the protection works against EMT transformed cells, as well.
Considering the deadly nature of melanomas, this is a very important information hopefully with future therapeutic development. In conclusion the manuscript has to be suggested for publication. In the meantime, before that several issues concerning the presentation of the manuscript has to address.

1. The English language of the text has to seriously edited as sometimes it is hardly understood, or easily misunderstood. etc: row 55 better: invasion of the tumor to the nearest tissue
Response: Text grammar and syntax has been edited, accordingly.
2. There are erroneous statements, such as cyclin D1 and D3 are the major player in melanoma, but although it is an important endpoint, nothing is mentioned about the major injuries in human melanomas, namely mutation of NRas, or BRaf, subsequently MEK, thus the MAPK pathway or sometimes beta catenin pathway.
3. row 659: cyclin D1 transcription is promoted by MAPK, or c-myc, but not by beta catenin pathway. Here active GSK3beta put on inactivating phosphorylation on the protein initiating its degradation. p21 is a cyclin dependent kinase inhibitor (and not kinase) transcribed by p53 or Foxo1 (unless she means p21 Ras), MMPs (like MMP1 is also transcribed by MAPk route (or NFkB or AKT) and not by the b catenin pathway.
Responses for 2 and 3: Thanks for pointing out these critical points. The roles of cyclins, NRas, BRaf, etc and beta-catenin have been inserted in the introductory text as well as in the respective paragraph of the discussion, accordingly.
4. row 65: hyaluronan -the statement of metabolism is OK but should be explained what is this about: glucose metabolism, UDP-glucose dehydrogenase upregulation -HAS2 upregulation – hyaluronic acid overexpression –EMT doi: 10.1038/s41388-019-0885-4.
Response: The text regarding the hyaluronan metabolism has been corrected given the details needed in the revised introduction, accordingly. The recommended reference: UDP-glucose 6- dehydrogenase regulates hyaluronic acid production and promotes breast cancer progression (by James M. Arnold et al), has also been added to guide the reader for further information (reference
number 14 of the revised Ms).
5. row 72: heparanase must be refer to its involvement in EMT
Response: It has been referred, accordingly.
6. row 98: the role of lumican, and its expression in young and old cartilage has nothing to do with the story
Response: It has been deleted as suggested.
7. row 128: here a few sentences are needed about the previous results with lumican, and describe what are the aims in this paper.
Response: New text, providing the requested information and the aim of this project, has been added in the indicated place of introduction, according to the suggestion of the reviewer.
8. row 301: cyclin D1 control sample must be shown
Response: Cyclin D1 in Lum+/+ and Lum-/- is now shown in figure 2, accordingly, and the respective changes made in the text.
As seen on Figure 1A, the lungs are almost free of nodules, therefore it is expected in the immunostaining of the lungs of figure 2A to have few nodules.
9. fig2 C I do not understand are there more nodules in the mock, triangles are individual animals? It should explain .
Response: Thanks for this point. The text has been corrected as to show it. Symbols presents the individual animals or mean values obtained from the IHC and these have now given in the legend, accordingly.
10. fig 3D: vinculin fluorescent pictures neither green nor blue is visible, the quality for presentation is better?
Response: Thanks for pointing it out. The picture in the text lost the quality/visibility. A much better quality has now been provided for the all-fluorescence images, accordingly, and it is now possible for the reader to see clearly the various immunostainings.
11. Fig 5B: Quantitation of MMP14 is wrong. No MMP14 is detected on the Western at snail-B16F1lumican, and the graph shows a considerable high column.
Response: Many thanks for this observation. Indeed, this figure was inserted by mistake and has nothing to do with MMP-14. It has now been corrected and the corresponding bars were given, accordingly.
12. Fig6: Total ERK- instead EPK, Mock B16 + lumican, the band is stronger than that of Mock B16 alone, grafic evaluation is the opposit.
Response: We changed EPK to ERK, thank you. Bands show that total ERK is decreased in mockcells + lumican and this is presented in the bars.
13. row 559: again, control cyclin D1 is needed (and western?)
Response: It has been inserted, accordingly in the figure 2A.
As seen on Figure 1A, the lungs are almost free of nodules, therefore it is expected in the immunostaining of the lungs of figure 2A to have few nodules.
Measurement was based in the histochemical images as described in the legend.
14. row 599 reference for heparanase and EMT is needed doi: 18632/oncotarget.26042
Response: Reference [43] to relate heparanase with EMT was added (Masola V, Bellin G, Vischini G, Dall'Olmo L, Granata S, Gambaro G, Lupo A, Onisto M, Zaza G. Inhibition of heparanase protects against chronic kidney dysfunction following ischemia/reperfusion injury. Oncotarget. 2018 Nov 16;9(90):36185-36201. doi: 10.18632/oncotarget.26324).
15. row 563: protrusions like microvesicles micrivesicles are not protrusions
Response: Thank you for this point too. It has been corrected, accordingly.
16. row 569: I would use immunostaining inseas of immunoblotting
Response: It has been corrected, accordingly.
17. row 574-579: the same is described 2 times
Response: It has been corrected, accordingly.
18. row 621: which activating phosphorylation of Akt was studied?
Response: The phosphorylation at phospho-AKT (ser473) was studied. It has been added in the results section, accordingly.
19. row 626: Without measuring I would not say, as final conclusion, that the real mechanism of lumican action is the inhibition of EGFR activation concluding to the dowregulation of FAK activity. Needless to say, it can considered as a potential mechanism, to study in the future.
Response: We agree with the proposal of the reviewer and the respective part has been modified, accordingly.

Round 2

Reviewer 1 Report

Summary: In the current revised version, Karamanou et al. have responded to many of our points. The introduction is improved. Organization and description of figures is improved.

However, several issues with figures remain. The descriptions of methods still are not complete or clear.

Major points:

  • Figure 3: This figure is now acceptable, but the corresponding methods section must be improved. Please add a complete description of the coating method.

  • Figure 4B: The absence of an appropriate loading control is a weakness. Thus, these data can only be interpreted qualitatively, not quantitatively. Please modify the text to remove quantitative references, e.g. line 487 slightly increased, line 489, 492 significantly reduced, 490 slight accumulation, 492 significantly decreased. Since actin is not an appropriate loading control, similar alterations must be made to figure 5 text: line 524, 526. (RT-qPCR data does not substitute for the loading control.)

  • Please provide whole immunoblots. The manuscript is not acceptable for publication without these.

  • Figure 5: This figure is improved; however, it would be preferable to have a different color for each bar in the graph. (Alternatively, label them as in Figure 6B.) Currently both Mock-B16F1 and Mock-B16F1 +Lum are red; Snail-B16F1 and Snail-B16F1 +Lum are both blue.
    In regard to panel c, please add the definition of control (which in the response to reviewer is listed as Mock) to the legend. An improved description of the invasion assay in the methods section is required. No description of cell starvation prior to seeding on collagen is currently given in the methods. Please clearly delineate sequence of events: when were cells starved, when was Lumican added, etc.

  • Figure 6A: Please indicate (provide a legend) what red and blue bars signify in Figure 6A.

  • Please include numbers of samples (biological and technical) in figure legends. In response to reviewers, it is stated that this has been added, but it is still missing in Figures 5,6.

Minor points:

  • Statements on lines 52-57 require references.

Reviewer 2 Report

The manuscript was modified according to the suggestion of reviewer.

Author Response

Thanks for your kind approval. Minor changes were made, accordingly.